# MicroRNA Expression Profile in TSC Cell Lines and the Impact of mTOR Inhibitor

**DOI:** 10.3390/ijms232214493

**Published:** 2022-11-21

**Authors:** Bartłomiej Pawlik, Szymon Grabia, Urszula Smyczyńska, Wojciech Fendler, Izabela Dróżdż, Ewa Liszewska, Jacek Jaworski, Katarzyna Kotulska, Sergiusz Jóźwiak, Wojciech Młynarski, Joanna Trelińska

**Affiliations:** 1Department of Pediatrics, Oncology and Hematology, Medical University of Lodz, Sporna 36/50, 91-738 Lodz, Poland; 2Postgraduate School of Molecular Medicine, Medical University of Warsaw, Żwirki i Wigury 81, 02-091 Warsaw, Poland; 3Department of Biostatistics and Translational Medicine, Medical University of Lodz, Mazowiecka 15, 92-215 Lodz, Poland; 4Department of Clinical and Laboratory Genetics, Medical University of Lodz, Pomorska 251, 92-231 Lodz, Poland; 5Laboratory of Molecular and Cellular Neurobiology, International Institute of Molecular and Cell Biology, Trojdena 4, 02-109 Warsaw, Poland; 6Department of Neurology & Epileptology and Pediatric Rehabilitation, The Children’s Memorial Health Institute, Dzieci Polskich 20, 04-730 Warsaw, Poland; 7The Children’s Memorial Health Institute, Dzieci Polskich 20, 04-730 Warsaw, Poland

**Keywords:** microRNA, mTOR inhibitor, rapamycin, tuberous sclerosis, shTSC1 and shTSC2 cells

## Abstract

The aim of this study was to assess the potential implication of microRNA on tuberous sclerosis (TSC) pathogenesis by performing microRNA profiling on cell lines silencing *TSC1* or *TSC2* genes using qPCR panels, before and after incubation with rapamycin. Significant differences in expression were observed between samples before and after rapamycin treatment in nineteen miRNAs in TSC1, five miRNAs in TSC2 and seven miRNAs in controls. Of miRNAs dysregulated before rapamycin treatment, three normalized after treatment in the TSC1 group (miR-21-3p, miR-433-3p, let-7g-3p) and one normalized in the TSC2 group (miR-1224-3p). Of the miRNAs dysregulated before rapamycin treatment in the *TSC1* and *TSC2* groups, two did not normalize after treatment (miR-33a-3p, miR-29a-3p). The results of the possible targets indicated that there are four common genes with seed regions susceptible to regulation by those miRNAs: *ZBTB20*, *PHACTR2*, *PLXNC1* and *ATP1B4.* Our data show no changes in mRNA expression of these targets after rapamycin treatment. In conclusion, results of our study indicate the involvement of miRNA dysregulation in the pathogenesis of TSC. Some of the miRNA might be used as markers of treatment efficacy and autonomic miRNA as a target for future therapy.

## 1. Introduction

Tuberous sclerosis (TSC) is a genetic disorder caused by mutation of the *TSC1* or *TSC2* genes, encoding two crucial proteins: TSC1 (hamartin) and TSC2 (tuberin). These proteins form the TSC1–TSC2 tumor suppressor complex, the mammalian target of rapamycin complex 1 (mTORC1) and mTORC2. mTORC1 comprises of mTOR, Raptor (regulatory-associated protein of mTOR), Deptor (DEP domain-containing mTOR-interacting protein), mLST8βL (mammalian lethal with SEC1 protein8/G-protein b-subunit-like protein), and PRAS40 (protein-rich Akt substrate of 40-kDa). mTORC2 is composed of mTOR, Rictor (rapamycin-insensitive companion of mTOR), mLST8 and mSin1 (mammalian stress-activated protein kinase-interacting protein 1) [1].

Hyperactive mTOR signaling in patients with TSC leads to abnormalities in many cellular processes, including cellular growth, proliferation, protein synthesis and metabolic control [2]. The TSC can result in severe neurological disorders, such as epilepsy, autism, mental retardation and subependymal giant cell astrocytoma (SEGA), as well as kidney involvement such as renal angiomyolipomas (AML) or renal cysts, pulmonary lymphangioleiomyomatosis (LAM) and cardiac rhabdomyomas [3].

Multiple clinical studies have now confirmed that mTOR inhibitors (everolimus and rapamycin) demonstrate clinical benefits against TSC tumors occurring in the kidney, brain and lungs [1,4,5,6,7].

However, clinical observations suggest that mTOR inhibitors do not completely eliminate symptoms of the disease. Partial response and tumor regrowth are often observed after treatment discontinuation [8,9]. It may indicate that hamartin and tuberin act not only as a complex, but in some ways may work independently of the mTOR pathway or that other factors may be involved in this process.

Recently, various studies showed that the AKT/mTOR pathway is also regulated by numerous epigenetic mechanisms, one of which could be microRNA (miRNA) inhibition. There is evidence that miR-19a/b and miR-130a/b inhibit mTORC1 in the epithelial cells of the kidney [10,11].

miRNAs target the mTOR pathway in several ways, either by interacting directly with mTOR itself or targeting key genes within the pathway, which ultimately affect mTOR function. These genes include upstream regulators of mTOR, such as *IGFR1*, *PI3K* and *AKT*, as well as negative regulators, such as *PTEN* [12].

Until now, however, few studies have been performed on the expression profiles of microRNAs present in the serum of patients with TSC. Our previous work compared miRNA profiling results from patients with TSC treated with sirolimus and everolimus, and 11 miRNAs were found to be dysregulated in the same directions following both treatments, with the most significant change of expression being observed for miR-136-5p, miR-376a-3p and miR-150-5p [13]. This indicates that both mTOR inhibitors may have homogenous effects on the mTOR pathway and allowed us to choose sirolimus (rapamycin) for the present experiment. However, the serum miRNA profile has been found to be dysregulated in TSC patients, which was only partially resolved after everolimus treatment [14]. Other studies demonstrated that rapamycin promotes the expression of several miRNA: miR-21 and miR-29b [15]. Those led us to the hypothesis that other signaling pathways are connected to mTOR by the microRNA network.

The aim of our study was to assess the potential influence of microRNA on TSC pathogenesis and alternative signaling pathways activations other than mTOR. To achieve this, microRNA profiling was performed on cell lines silencing *TSC1* or *TSC2* genes before and after incubation with rapamycin.

## 2. Results

From the 752 miRNAs tested in the cell-conditioned media, 318 were detected in at least one sample in each group before and after rapamycin treatment (Appendix A). Six miRNAs (hsa-let-7i-5p, hsa-let-7g-3p, hsa-miR-29a-3p, hsa-miR-433-3p, hsa-miR-33a-3p and hsa-miR-21-3p) demonstrated significantly different expression between the untreated TSC1 and control (Figure 1A); three miRNAs (hsa-miR-29a-3p, hsa-miR-515-5p and hsa-miR-1224-3p) differed between the untreated TSC2 and control (Figure 1B) and two miRNAs (hsa-let-7g-3p and hsa-miR-515-5p) differed between the untreated TSC1 vs. TSC2 groups (Figure 1C). The overlap of these differences in expression is shown in Figure 2A.

Within the group after rapamycin treatment, six miRNAs (hsa-miR-194-5p, hsa-miR-29a-3p, hsa-miR-454-5p, hsa-33a-3p, hsa-515-5p and hsa-miR-208a-3p) were differentially expressed between TSC1 and control, with two miRNAs (hsa-miR-29a-3p and hsa-miR-194-5p) demonstrating FDR < 0.15 (Figure 1D); three miRNAs (hsa-miR-129-2-3p, hsa-miR-29a-3p and hsa-miR-515-5p) were differentially expressed between TSC2 and control (Figure 1E) and one miRNA (hsa-miR-570-3p) between TSC1 and TSC2 groups (Figure 1F). The overlap of these differences in expression is shown in Figure 2B.

In addition, nineteen miRNAs in TSC1, five miRNAs in TSC2 and seven miRNAs in the control demonstrated significant differences in expression before and after rapamycin treatment (Figure 1G–I and Figure 2D). The overlap of those differences in expression is shown in Figure 2C.

The effect of treatment on miRNA expression in the TSC1 and TSC2 groups is shown in Figure 3. From six miRNAs dysregulated before rapamycin treatment in TSC1, three normalized after treatment (miR-21-3p, miR-433-3p, let-7g-3p; Figure 3C–E) and the other two did not normalize after treatment (miR-33a-3p, miR-29a-3p; Figure 3G,H). Of the three miRNAs dysregulated before rapamycin incubation in the TSC2 group, miR-1224-3p normalized after treatment (Figure 3F) and miR-29a-3p did not (Figure 3H).

To evaluate the implications of the miRNAs which did not normalize after treatment, the experimentally validated targets from miRTarBase were analyzed using miRWalk software (http://mirwalk.umm.uni-heidelberg.de/, accessed on 21 October 2022). In total, 51 genes were found; however, none of them were targeted by both miR-33a-3p and miR-29a-3p (Figure 4, Appendix A), and no Reactome or KEGG pathways involving the two miRNAs were identified (Appendix A). Subsequent analysis revealed putative targets (predictions from miRDB generated by MirTarget) for the selected miRNAs (Appendix A).

Four genes were predicted to be potentially regulated by both miR-29a-3p and miR-33a-3p: zinc finger and BTB domain-containing 20 (ZBTB20), plexin C1 (PLXNC1), phosphatase and actin regulator 2 (PHACTR2) and ATPase Na^+^/K^+^ transporting family member Beta 4 (ATP1B4). Further qPCR analysis of mRNA expression level of these genes was performed to validate the in silico results.

Rapamycin treatment did not affect *ZBTB20* expression in *TSC1* defected cells (*p* = 0.4995); however, slightly higher levels were observed in sh*TSC2* cells (*p* = 0.0419). Rapamycin did not affect the expression of *PHACTR2* (*p* = 0.8335 for shTSC1 and *p* = 0.8139 for shTSC2) or *PLXNC1* (*p* = 0.6035 for shTSC1 and *p* = 0.6029 for shTSC2) (Figure 5). The expression level of *ATP1B4* gene was below the detection threshold.

To evaluate the potential transcription factors which may regulate miR-33a and miR-29a expression, TFBS (transcription factor binding site) analysis from Transfac was performed using PROMO (Alggen) software (https://alggen.lsi.upc.es/) [16]. Twenty-seven transcription factors were predicted in two input sequences for miR-33a and miR-29a within a dissimilarity margin less than or equal to 5%: C/EBPβ, TBP, RXR-α, GR-β, TFIID, GR-α, AP-2αA, c-Ets-2, FOXP3, YY1, Pax-5, p53, GATA-1 STAT4, c-Ets-1, XBP-1, ER-α, TFII-I, T3R-β1, GR, VDR, C/EBPα, c-Jun, NF-Y, PR B and PR A (Appendix A).

## 3. Discussion

TSC results from the presence of rare defects in the *TSC1* or *TSC2* genes, which lead to dysregulation of the mTOR signaling pathway and uncontrolled cell growth and proliferation. Patients with those mutations exhibit multi-organ manifestations, of which the highest morbidity is associated with neurological disorders. Preclinical and clinical studies indicate the use of mTOR selective inhibitors in the treatment of TSC [17,18,19].

In this work, the in vitro model of TSC was established by lentiviral transduction of shRNA for either *TSC1* or *TSC2* genes in neural stem cells. The cells were then treated with 20 nM of rapamycin to reflect the clinical effect of mTOR inhibitors, rapamycin being the most commonly used drug in TSC treatment [6].

The main result of our study was the demonstration of miRNA dysregulation in cell lines silencing TSC1 or TSC2 genes. Incubation with rapamycin caused normalization of miR-21-3p, miR-433-3p and let-7q-3p in the TSC1 group and miR-1224-3p in the TSC2 group. However, some of the miRNA did not normalize after rapamycin treatment: miR-33a-3p and miR-29-3p in the TSC1 group and miR-29a-3p in the TSC2 group.

While mTOR inhibitors have been found to be effective at controlling the major symptoms of TSC, they usually demonstrate only a partial response [6,17,20]. It was previously reported that during TSC development, the serum miRNA profile is altered in an mTOR-dependent manner, since the inhibition of this pathway partially negates the miRNA disorder [13,14]. Upregulation of some miRNA and normalization after rapamycin treatment corresponds to the therapeutic response to rapamycin in TSC; however, miRNAs which did not normalize may be responsible for the partial efficacy of the therapy.

Therefore, the present study examines the miRNA profile from cell-conditioned media of TSC in vitro. In the *TSC1*-defective cells, levels of miR-21-3p, miR-443-3p and let-7g-3p were altered; however, these were normalized following rapamycin treatment. In the *TSC2*-defective cells, the level of miR-1224-3p was normalized during treatment. Similarly, Calsina et al. found the level of miR-21-3p miRNA to correlate with TSC2 expression and mTOR activation, suggesting that it could be a potential stratification biomarker for mTOR inhibitor treatment among patients with pheochromocytomal and paragangliomal neuroendocrine tumors [21]. The link between the mTOR pathway and miR-1224 has also been reported in osteosarcoma cells [22]. These results are in line with our present findings and suggest that miRNAs could be considered as biomarkers of mTOR inhibitor treatment efficacy in TSC.

Two other miRNAs did not normalize after rapamycin treatment in the present in vitro model of TSC. A defect in the *TSC1* gene results in higher levels of miR-33a-3p and miR-29a-3p compared to wild type cells, with a remarkable boost of their expression after rapamycin treatment. A similar effect was observed in *TSC2*-defective cells for miR-29a-3p, but not 33a-3p. Literature data about miR-33a indicate that it participates in nine signaling pathways that are associated with cancer, one of which is mTOR [23]. In the TSC cells, decreased miR-33 expression was reported after rapamycin treatment [24].

In breast cancer cells, overexpressed miR-29a-3p promotes the repression of HIF1α (hypoxia-inducible factor 1 subunit alpha) protein, the downstream mTOR effector [25,26]. Upregulated miR-29-3p could inhibit hepatocellular carcinoma cell proliferation, migration, invasion, and metastasis through inhibition of Robo1 and inactivation of the PI3K/Akt/mTOR signaling pathway [27].

However, this previous data contradicts our present findings. This may be explained by the possible role of these miRNAs in the regulation of alternative, mTOR-independent signaling pathways during TSC development.

In silico analysis was performed to identify the targets of miR-33a-3p and miR-29a-3p; although 51 genes were found in miRTarBase, none of them were targeted by both miRs. Further MiRTarget analysis revealed four potential targets: *ATP1B4*, *ZBTB20*, *PHACTR2* and *PLXNC1*. The *ATP1B4* gene encodes the Na^+^/K^+^ ATPase plasma membrane protein, crucial for electrochemical gradient maintaining. This gene is highly expressed in thyroid and skeletal muscle [28]. In these experiments, the expression of *ATP1B4* in nerve cells was found to be below the detection threshold. *ZBTB20* encodes the transcription factor Zbtb20, which is highly expressed by neural progenitor cells in forebrain neurogenic niche [29]. This protein was previously described to be crucial in hippocampal specification and regulation of neurogenesis [30].

The regulation of transmembrane receptor Plexin C1 (encoded by *PLXNC1*) is crucial for direct topographic dopaminergic circuit formation [31]. Plexins are also expressed in the immune cells, and Plexin C1 activity is involved in acute inflammation and the attenuated migration and mobility of dendritic cells [32,33].

The exact function of *PHACTR2* remains unknown. Its protein product may have actin binding and phosphatase inhibition activity [34]. Its activity is implicated in Parkinson’s disease, Alzheimer’s disease and multiple sclerosis [35,36], and some evidence exists that PHACTR2 may influence the development of cancer [37,38]. Apart from tissue specificity, the implication of the activity of these genes in TSC remains unknown.

Our present findings indicate that in the TSC in vitro model, all the above genes have miR-33a-3p and miR-29a-3p binding sites. Moreover, qPCR analysis confirmed that expression of *ZBTB20*, *PHACTR2* and *PLXNC1* did not alter after rapamycin treatment, which may indicate that activation of these miRs and their function are independent of the mTOR signaling pathway.

The results of the PROMO analysis revealed the 27 transcription factors which have the binding motifs in both miR-33a and miR-29a sequences. Although these TFs have no experimentally validated data regarding their relationship with miR-33a and miR-29a, some are regulated indirectly by mTOR signaling. Yang et al. report that IL-1β expression in macrophages is regulated by TSC1 protein via the mTOR dependent C/EBPβ pathway [39]. Similar results indicate that stimulating the C/EBPβ pathway by mTOR promotes tumor growth and immune suppression [40]. Moreover, increased phosphorylation of TSC2 protein and inhibition of mTOR by methamphetamine led to increased C/EBPβ expression and thus activation of neuronal autophagy [41].

Other studies indicate that constitutive mTOR signaling activation occurring upon TSC1 or TSC2 defects leads to accumulation of p53 in TSC tumors, and the efficacy of mTOR inhibitors may depend on p53 status [42]. In general, inhibition of mTOR results in temporary phosphorylation of p53 protein and p53 could in turn regulate mTOR activity by AMP kinase and the TSC1/TSC2 complex [43].

Therefore, these findings suggest that miRNA may play a pivotal role in TSC pathogenesis, and that some may serve as biomarkers of treatment efficacy. The autonomy of miR-29a-3p and miR-33a-3p observed after rapamycin treatment, and the lack of changes in their downstream regulatory targets suggests that they might be mTOR independent. One explanation could be that other signaling pathways are connected to mTOR by the microRNA network. The most promising crosstalk signaling pathways may involve C/EBPβ or p53 transcription factors; however, further studies are needed. Additionally, miR-29a-3p and miR-33a-3p could serve as a targeted therapy for tumors associated with TSC not responding to rapamycin treatment.

## 4. Materials and Methods

### 4.1. Cell Culture

The skin biopsy from a healthy volunteer was obtained with informed and written consent and processed anonymously. The study was approved by the Ethics Committee of the Children’s Memorial Health Institute, Warsaw, Poland (decision no. 112/KBE/2013). All experimental procedures/methods were performed in accordance with the relevant guidelines and regulations.

Induced pluripotent stem cells were obtained as described previously by Liszewska et al., 2021; briefly, human fibroblasts derived from skin biopsy (male, 40 years old) were cultured in Dulbecco’s modified Eagle’s medium (DMEM): high glucose medium supplemented with 10% fetal bovine serum, 1% penicillin-streptomycin (Sigma-Aldrich, St. Louis, MO, USA). The culture was maintained at 37 °C in a humidified atmosphere with 5% CO_2_. At passage 4, fibroblasts were seeded at a density of 1 × 10^4^ cells/cm^2^, and on the two following days were transduced with lentivirus carrying Oct4, Klf4, Sox2 and c-Myc in the presence of 5 ug/mL Polybrene (Sigma-Aldrich, St. Louis, MO, USA). Two days after transduction, the fibroblasts were transferred onto mouse embryonic fibroblasts (MEFs) (EmbryoMax Primary Mouse Embryonic Fibroblasts—PMEF-CFL, Sigma-Aldrich, St. Louis, MO, USA) inactivated with Mitomycin C (Sigma-Aldrich, St. Louis, MO, USA) and cultured in iPSC medium composed of DMEM/F12, 20% Knockout Serum Replacement, 1% non-essential amino acid stock, 1 mM GlutaMax, 1 mM Sodium pyruvate, 100 μM β-mercaptoethanol, 1% penicillin-streptomycin (ThermoFisher Scientific, Waltham, MA, USA) and 10 ng/mL bFGF (Alomone, Jerusalem, Israel), with medium change every other day. After three weeks, iPSC colonies were manually picked and expanded. Cells were passaged manually every 4–5 days. Undifferentiated iPSC colonies were collected, dispersed by pipetting and small clumps of cells were plated onto fresh mitotically inactivated MEFs in iPSC medium. Following this, the iPSCs were differentiated to neural stem cells (NSC) applying the embryoid body-based protocol. Then, iPSC colonies were detached using collagenase (Sigma-Aldrich, St. Louis, MO, USA), washed with phosphate-buffered saline (PBS) and transferred to a low attachment culture dish [44].

To form embryoid bodies (EBs), the cells were cultured in suspension in iPSC medium without bFGF for five days, with half medium changes every other day. The EBs were then attached onto Matrigel (Corning) coated plates and cultured in NES medium composed of DMEM/F12, N2 supplement (1:100), B27 (1:100), 1 mM GlutaMax, 1% penicillin-streptomycin (ThermoFisher Scientific, Waltham, MA, USA), 20 ng/mL bFGF and 20 ng/mL EGF (Alomone, Jerusalem, Israel). After four days, the emerging neural rosette structures were manually isolated, dispersed by pipetting and plated on Matrigel in NSC medium.

Upon reaching confluency, NES cultures were passaged every 4–5 days using Accutase (Sigma-Aldrich, St. Louis, MO, USA) for further expansion. The NES were then transduced with lentivirus: control or carrying shRNA (short hairpin RNA) for the TSC1 or TSC2 genes (TSC1 shRNA ”1” GAAGAAGCTGCAATATCTA; TSC1 shRNA “2” GGGAGGTCAACGAGCTCTATT; TSC2 shRNA “1” CACTGGCCTTGGACGGTATTG; TSC2 shRNA “2” GGATTACCCTTCCAACGAA). Additionally, the virus had a puromycin-resistant cassette. Forty-eight hours after transduction, 1 ug/mL puromycin (Sigma-Aldrich, St. Louis, MO, USA) was added to the culture medium to select cells that incorporated the virus. To exclude the site-specific effect of the interfered shRNA, two different shRNAs were used for each analyzed gene.

For the rapamycin treatment experiments, the cells were seeded at a density of 5 × 10^4^/cm^2^ in 12-well dishes in NES medium. The next day, shTSC1 and shTSC2 cells were treated with 20 nM rapamycin (Tocris, Bristol, UK) and the controls were not. After 48 h, the culture media were transferred to RLT buffer (Qiagen, Hilden, Germany) and cells were collected in RLT buffer (Qiagen, Hilden, Germany). All samples were stored at −80 °C for further analysis. Both the treated and untreated groups included two shTSC1, two shTSC2 and one control sample.

### 4.2. MicroRNA Profiling

Prior to miRNA extraction, exosomes were isolated from cell-conditioned media using the miRCURY Exosome Cell/Urine/CSF Kit (Qiagen, Hilden, Germany), and miRNA isolation was performed using the miRCURY RNA Isolation Kit (Qiagen, Hilden, Germany) according to the manufacturer’s protocol. To allow for normalization of sample-to-sample variation in the RNA isolation, prior to purification, each sample was spiked with UniSp2, UniSp4, UniSp5, each at a different concentration in 100-fold increments, with UniSp6 and cel-miR-39-3p as positive controls for cDNA synthesis. RNA quality was determined with the Agilent RNA 6000 Nano Kits using 2100 Bioanalyzer (Agilent Technologies, St. Clara, CA, USA). The degradation rate of total RNA was determined using RIN values. Only the samples with RIN > 7 were included in further analysis. Directly after isolation, RNA was subjected to reverse transcription [45].

### 4.3. Reverse Transcriptase Reaction

MiRCURY LNA Reverse Transcription Kit and the stem-loop RT primers for miRNAs were purchased from Qiagen (Hilden, Germany) and used to synthesize cDNA according to the guidelines provided by the manufacturer. Ten nanograms of total RNA were added to the reaction tube to make up a final volume of 10 μL reaction mix. The mix was then incubated (60 min, 42 °C and 5 min, 95 °C) in the thermal cycler [45].

### 4.4. Real-Time PCR (qPCR)

The expression of 752 human miRNAs was quantified in all samples. Real-time PCR was performed on a LightCycler480 (Roche, Basel, Switzerland) instrument using miRCURY LNA miRNome Human PCR Panel I and II (Qiagen, Hilden, Germany). The reaction was performed at 95 °C for 2 min, followed by 45 amplification cycles at 95 °C for 10 s and 56 °C for 1 min. Then, 1 ng of cDNA was used in each PCR reaction and all samples were amplified simultaneously in duplicate in a single run.

### 4.5. RNA Isolation and Gene Expression Study

Total RNA was extracted from samples stored in RLT buffer by using the RNeasy Mini Kit (Qiagen, Hilden, Germany), according to the manufacturer’s protocol. The DNA contamination was removed by DNA digestion, using RNase-FreeDNase Set (Qiagen, Hilden, Germany). RNA concentration was determined using a NanoDrop 8000 Spectrophotometer (Life Technologies, Carlsbad, CA, USA). For cDNA preparation, RNA was reverse transcribed using the High-Capacity cDNA Reverse Transcription Kit (Life Technologies, Carlsbad, CA, USA) for 10 min at 25 °C, 120 min at 37 °C and 5 min at 85 °C. TaqMan Gene Expression assays (Applied Biosystems, Carlsbad, CA, USA) were used for real-time qPCR measurement of *ZBTB20* (cat.no. Hs00210321), *ATP1B4* (cat.no. Hs00201320), *PHACTR2* (cat.no. Hs01112117) and *PLXNC1* mRNA expression (cat.no. Hs00194968). *GAPDH* (cat.no. Hs03929097) was selected as a control. The conditions on the thermocycler were as follows: 50 °C for 2 min, 95 °C for 10 min, followed by 40 cycles of 95 °C for 15 sec and 60 °C for 1 min. All samples were tested in duplicate and the 2^−ΔΔCq^ value was calculated [46] with *GAPDH* as a reference gene. The relative mRNA expression was used for further comparisons.

### 4.6. Bioinformatical and Statistical Analysis

Before statistical analysis, the data were filtered: spike-ins were removed and Cq values above 37 were marked as undetects, according to the panel manufacturers’ recommendation. Following this, miRNA expression was normalized by subtracting it from the averaged Cq of hsa-let-7a-5p, hsa-miR-19a-3p and hsa-miR-20a-5p; a higher resulting value indicates higher expression. This combination of three reference miRNAs was found by NormiRazor [47] for our recent study on miRNA expression in the serum of TSC patients [13]. This normalization strategy enables the most adequate comparison between serum and cell lines. Following this, miRNAs that were not expressed in at least one sample in each group (control, shTSC1, shTSC2) before and after rapamycin treatment were removed. If expression was detected in both duplicates, the two values were averaged, and if expression was detected only in one sample, this value was taken for further analysis.

Differential expression analysis was performed using limma [48] with a model that included group, treatment status and an additional sample to account for repeated measurements. Any miRNAs that were differentially expressed between groups were identified based on this model, as well as any changes in expression attributable to rapamycin. Differences in expression were considered significant when the *p*-value was below 0.01 and the fold-change (FC) above 2 or below 0.5. Any binding sites for selected miRNAs whose expression changed in samples with shTSC1/2 and did not normalize after rapamycin treatment was subjected to in silico analysis in miRWalk [49]. Experimentally validated targets from miRTarBase [50] were then identified, and binding sites predicted by MirTarget (miRDB) [51].

In silico analysis of predicted transcription factor binding sites (TFBS) in DNA sequences of miR-33a (chr22:41899944–41901012) and miR-29a (chr7:130875747–130876810) was performed in PROMO (Alggen) [24].

The mRNA expression level of the resulting target genes, which were experimentally validated later, were compared between the groups using the unpaired Student’s *t*-test for comparisons between one shTSC1 or TSC2 before and after rapamycin treatment.

## 5. Conclusions

The dysregulations of miRNAs found in cell lines silencing TSC1 or TSC2 genes before and after rapamycin treatment indicate their involvement in TSC pathogenesis. The autonomic miRNA (miR-29a-3p and miR-33a-3p) may represent an attractive target for preclinical studies in TSC-related tumors in the future. Further studies are needed to confirm link between C/EBPβ and p53 transcription factors and autonomic miRNA.

## Figures and Tables

**Figure 1 ijms-23-14493-f001:**
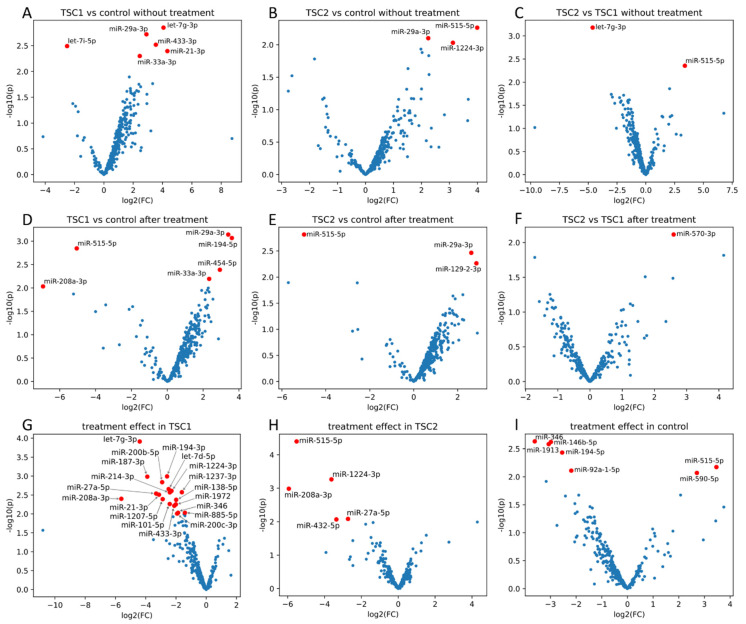
Differences in miRNA expression between TSC1 defect and control (**A**), TSC2 defect and control (**B**), and TSC1 and TSC2 defect (**C**), without rapamycin treatment and after rapamycin treatment (**D**, **E**, **F**, respectively). Effect of rapamycin treatment on miRnome in TSC1 (**G**), TSC2 (**H**) and control (**I**). All fold changes (FC) and *p*-values from limma (Appendix A). Red marksdenote miRNAs with *p* values <0.01.

**Figure 2 ijms-23-14493-f002:**
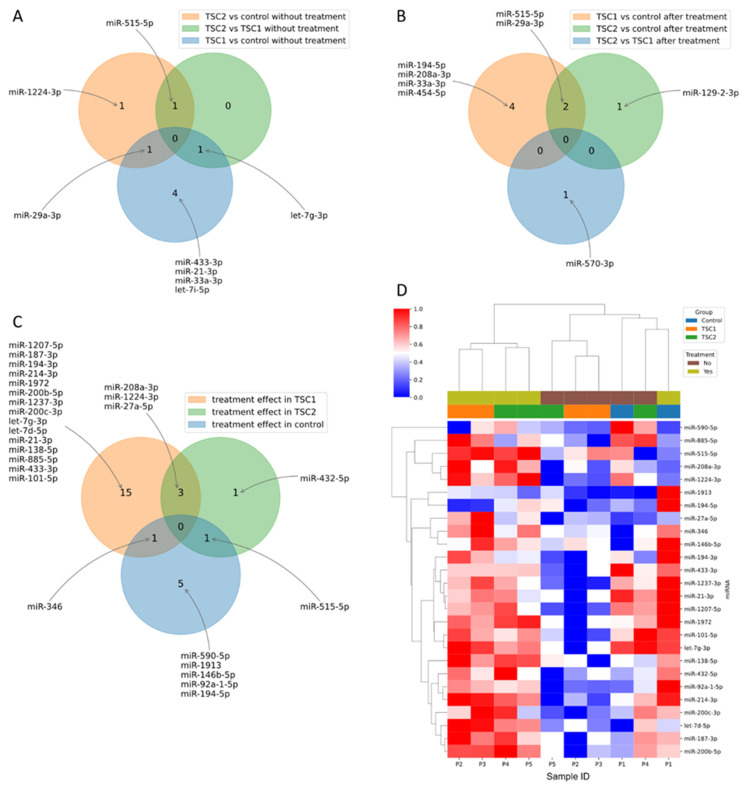
Overlap of differentially expressed miRNAs before (**A**) and after (**B**) rapamycin treatment. Overlap of treatment effects in *TSC1* defect, *TSC2* defect and control (**C**). Expression of miRNAs changed during rapamycin treatment in any of the groups (**D**).

**Figure 3 ijms-23-14493-f003:**
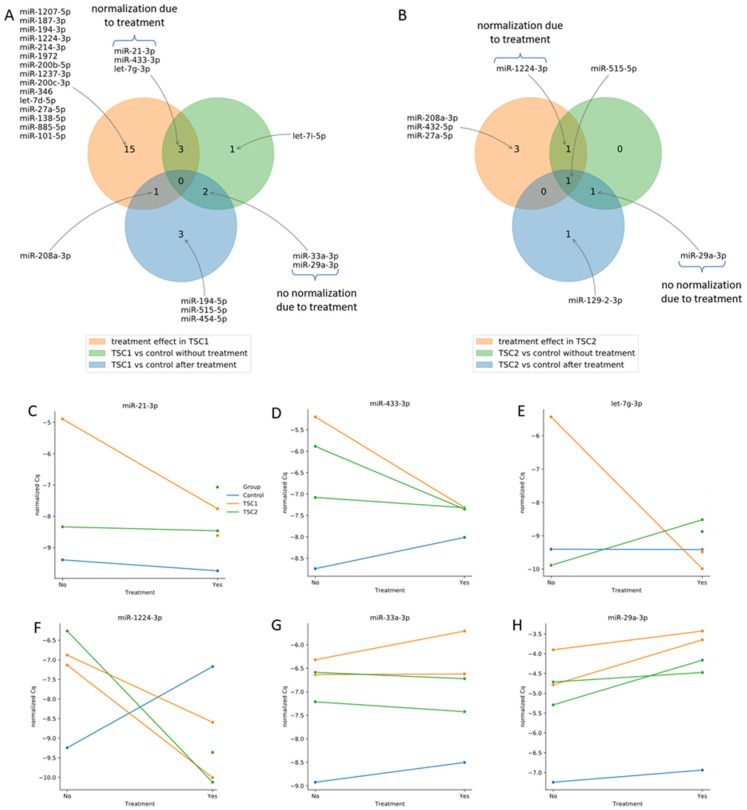
Effect of rapamycin treatment in *TSC1* (**A**) and *TSC2* (**B**) defects on miRnome. Expression of miRNAs that normalized after rapamycin treatment: miR-21-3p (**C**), miR-433-3p (**D**), let-7g-3p (**E**), miR-1224-3p (**F**) and those that did not normalize: miR-33a-3p (**G**) and miR-29a-3p (**H**).

**Figure 4 ijms-23-14493-f004:**
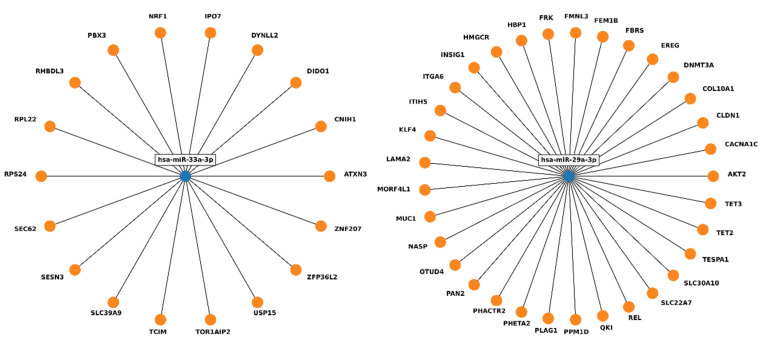
Experimentally validated targets of miR-33a-3p and miR-29a-3p, found by miRWalk in miRTarBase.

**Figure 5 ijms-23-14493-f005:**
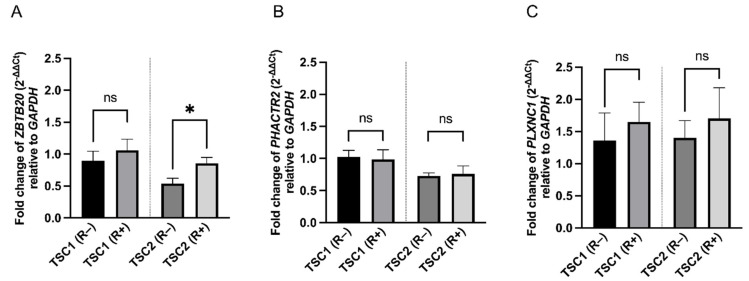
Effect of rapamycin treatment on *ZBTB20* (**A**), *PHACTR2* (**B**) and *PLXNC1* (**C**) genes in *TSC1* or *TSC2* defected cells. Asterisks represent statistical significance from an unpaired Student’s *t*-test (* *p* < 0.05), *ns* represents no statistical significance. Error bars represent SEM values.

## Data Availability

All microRNA data analyzed during the study are included in Appendix A. All other data generated during the study are available from the corresponding author upon reasonable request.

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
