# Peer review of "MicroRNA Expression Profile in TSC Cell Lines and the Impact of mTOR Inhibitor"

_ijms, 2022, doi:10.3390/ijms232214493_

Round 1
Reviewer 1 Report
Dear authors, I read with interest your work and found it very interesting.
Some discussion concerning your previous works on patients in treatment with other mTOR inhibitors is missing. This should be included in the discussion. Why did you choose this inhibitor when you already had an idea of the effects of the other mTOR inhibitors in vivo? I would have expected to see if in vitro studies could represent in vivo responses to treatment.
I need some clarification on the way the normalization was done. Did you use the geometrical mean of normalizers? Why didn't you use the common use 2delta,delta Ct? Would the answer be different?
The selection of normalizers and the normalization method are critical steps in microRNA expression profiles.
Normalyzer must have special characteristics: "the miRNA must be highly expressed in most, if not all, of the samples; (2) the miRNA must be consistently expressed, (3) only one representative from a given miRNA family (closer)should be considered.
Do the normalizers you choose have these characteristics?
"miRNA expression was normalized by subtracting it from the 364 averaged Cq of hsa-let-7a-5p, hsa-miR-19a-3p and hsa-miR-20a-5p;"
This type of normalization is very poor for microRNA expression analysis
I think you should see if the results are consistent using another type of normalization method/calculation.
https://pubmed.ncbi.nlm.nih.gov/32993488/
https://www.researchgate.net/publication/24015723_
https://journals.plos.org/plosone/article?id=10.1371/journal.pone.0193173
Author Response
Dear Reviewer,
We greatly appreciate your interest in our manuscript and made our best effort to introduce the changes suggested during Your revision. The responses to each of Your concerns are listed below. We hope that the revised manuscript will prove to be of sufficient quality to consider its publication in your prestigious journal.
Joanna Trelinska M.D., Ph.D.
Manuscript ID: ijms-2016560 
Reviewer comments:
Reviewer 1
Comments and Suggestions for Authors
Dear authors, I read with interest your work and found it very interesting.
- Some discussion concerning your previous works on patients in treatment with other mTOR inhibitors is missing. This should be included in the discussion.
Response:
Some discussion concerning our previous work on patients treated with mTOR inhibitor is already included in the text:
It was previously reported that during TSC development, the serum miRNA profile is altered in an mTOR-dependent manner, since the inhibition of this pathway partially negates the miRNA disorder (Trelinska et al. 2016; Pawlik et al. 2022b).
We have added one more sentence:
“Upregulation of some miRNA and normalization after rapamycin treatment correspond to the therapeutic response to rapamycin in TSC, however miRNAs which did not normalize may be responsible for the partial efficacy of the therapy.”
- Why did you choose this inhibitor when you already had an idea of the effects of the other mTOR inhibitors in vivo?
Response:
Our previous work (Pawlik et al. 2022) have compared miRNA profiling results from patients with TSC treated with sirolimus and everolimus and 11 miRNAs were found to be dysregulated in the same directions following both treatments, with the most significant change of expression being observed for miR-136-5p, miR-376a-3p and miR-150-5p. This indicates that both mTOR inhibitors may have homogenous effects on the mTOR pathway and allowed us to choose sirolimus (rapamycin) for present experiment. Additionally, rapamycin is a most studied from mTOR inhibitor in cellular experiments.
- I would have expected to see if in vitro studies could represent in vivo responses to treatment.
Response:
Unfortunately our present in vitro study did not overlap our previous in vivo results of changes in miRNA profiling in context to treatment response.
- I need some clarification on the way the normalization was done. Did you use the geometrical mean of normalizers? Why didn't you use the common use 2delta,delta Ct? Would the answer be different?
The selection of normalizers and the normalization method are critical steps in microRNA expression profiles.
Normalyzer must have special characteristics: "the miRNA must be highly expressed in most, if not all, of the samples; (2) the miRNA must be consistently expressed, (3) only one representative from a given miRNA family (closer)should be considered.
Do the normalizers you choose have these characteristics?
"miRNA expression was normalized by subtracting it from the 364 averaged Cq of hsa-let-7a-5p, hsa-miR-19a-3p and hsa-miR-20a-5p;"
This type of normalization is very poor for microRNA expression analysis
I think you should see if the results are consistent using another type of normalization method/calculation.
https://pubmed.ncbi.nlm.nih.gov/32993488/
https://www.researchgate.net/publication/24015723_
https://journals.plos.org/plosone/article?id=10.1371/journal.pone.0193173
Response:
Thank you for pointing out the importance of a suitable reference gene for expression measurements. We are aware of the fact intimately and have struggled with a proper selection of those in this and previous project. The standard ddCt method does not work for serum for two reasons. Firstly the first delta is Ct(ref)-Ct(gene) (which is possible and we apply this), the second d is dCt(baseline)-dCt(experiment) which in the case of unpaired controls and cases like in our study is impossible to perform. Therefore, as stated in the original paper by Livak and Schmittgen, we rely on dCt for expression comparisons between the two groups. The dCts are internally standardized per each patient and thus comparable as ratios of gene of interest / reference (https://pubmed.ncbi.nlm.nih.gov/11846609/).
For the choice of references we used NormiRazor (line 367) and the output recommended this combination of miRNAs. This tool includes GeNorm (and other tools) to produce an integrated stability score for combinations of reference genes. In biofluids there is no standard reference gene like in tissue samples where one can rely on housekeeping genes (ACTB, GAPDH or U6), the use of which is discouraged for biofluids as stated in the 3rd publication you mention. Thus, a combination of stable reference miRs is the safest option for relative quantification. Some of the authors of this work are actually authors of the NormiRazor publication (Grabia, Smyczynska, Fendler) and were consulted on the choice of reference genes during data analysis.

Reviewer 2 Report
In this work the authors investigated the role of microRNA on tuberous sclerosis pathogenesis by performing microRNA profiling on cell lines silencing TSC1 or TSC2 genes, before and after incubation with rapamycin. The topic of this work is interesting and the methods are sound.
However, I have different major points to address:
1. The authors should better explain at the end of the introduction section the originality and the added value of their work. It is important to underline what is known in literature and what the authors want to add as new and impactful prospective on this topic and on their work. For instance, the choice of rapamycin is not clear to me. They wrote the it is an inhibitor of mTOR signalling but then they also reported some limitations. Similarly, in the abstract section where it is not clear why the microRNA will be detected before and after rapamycin treatment. I think that the use of rapamycin has not to be introduced in the discussion section.
2. What do the authors mean with “untreated TSC1 and control (for instance line 94)”? what is the difference between “untreated” and “control”?
3. The discussion should start with a summary of the main results obtained.
4. the link between the sentences from lines 178-180 to lines 180-183 is not clear to me. Moreover, the reference to the Calsina et al. 2019’s work is not clear.
5. In general, the authors should be clearer and incisive in the whole discussion section, better comparison with other authors and works, and they should add a limitation section.
6. The conclusions should be much more incisive and they should look at future directions mainly in relation to translational level. This paragraph: “The variability of miR-29a-3p and miR-33a-3p after rapamycin treatment and the lack of changes in their downstream regulatory targets suggests that they might be mTOR independent.” (lines 392-393, lines 40-41) is not clear.
The abstract, the end of the discussion and the conclusions are the same words.
Minor points:
1. Check for genes: they must be written in italics (for instance lines 77-78, 16, 164,165).
2. Check for English errors, for instance “The variability of miR-29a-3p and miR-33a-3p after rapamycin treatment and the lack of changes in their downstream regulatory targets suggests (suggest) that they might be mTOR independent.” (lines 392-393, lines 40-41); “Another two miRNAs…” (line 195).
3. There are two repetitions in the abstract section: “data” in lines 38-39.
4. Check for typos, for instance, eliminate point at line 126, insert point at line 270. Check lines 204, 208-209.
5. There is a difference in font size at the lines 263-268.
Author Response
Dear Reviewer,
We greatly appreciate your interest in our manuscript and made our best effort to introduce the changes suggested during Your revision. The responses to each of Your concerns are listed below. We hope that the revised manuscript will prove to be of sufficient quality to consider its publication in your prestigious journal.
Joanna Trelinska M.D., Ph.D.
Manuscript ID: ijms-2016560 
Reviewer comments:
Reviewer 2
Comments and Suggestions for Authors
In this work the authors investigated the role of microRNA on tuberous sclerosis pathogenesis by performing microRNA profiling on cell lines silencing TSC1 or TSC2 genes, before and after incubation with rapamycin. The topic of this work is interesting and the methods are sound.
However, I have different major points to address:
- The authors should better explain at the end of the introduction section the originality and the added value of their work. It is important to underline what is known in literature and what the authors want to add as new and impactful prospective on this topic and on their work. For instance, the choice of rapamycin is not clear to me. They wrote the it is an inhibitor of mTOR signalling but then they also reported some limitations.
Response:
According to reviewer suggestion we have rewritten this paragraph
Our previous work (Pawlik et al. 2022) have compared miRNA profiling results from patients with TSC treated with sirolimus and everolimus and 11 miRNAs were found to be dysregulated in the same directions following both treatments, with the most significant change of expression being observed for miR-136-5p, miR-376a-3p and miR-150-5p. This indicates that both mTOR inhibitors may have homogenous effects on the mTOR pathway and allowed us to choose sirolimus (rapamycin) for present experiment. However, the serum miRNA profile has been found to be dysregulated in TSC patients, which was only partially resolved after everolimus treatment (Trelinska et al. 2016). Other studies demonstrated that rapamycin promotes the expression of several miRNA: miR-21 and miR-29b (Trindade et al. 2013; Liu et al. 2019). Those led us to hypothesis that other signaling pathways are connected to mTOR by the microRNA network.
- Similarly, in the abstract section where it is not clear why the microRNA will be detected before and after rapamycin treatment.
Response:
In the abstract section we stated that:
“Significant differences in expression were observed between samples before and after rapamycin treatment in 19 miRNAs in TSC1, five miRNAs in TSC2 and seven miRNAs in controls.”
It means that miRNA were detected before and after rapamycin treatment, however the expression was different.
- I think that the use of rapamycin has not to be introduced in the discussion section.
Response:
The use of rapamycin was introduced in the discussion section:
“In this work, the in vitro model of TSC was established by lentiviral transduction of shRNA for either TSC1 or TSC2 genes in neural stem cells. The cells were then treated with 20 nM of rapamycin to reflect the clinical effect of mTOR inhibitors, rapamycin being the most commonly-used drug in TSC treatment (Bissler et al. 2008).”
- What do the authors mean with “untreated TSC1 and control (for instance line 94)”? what is the difference between “untreated” and “control”?
Response:
The experimental setup was divided into two main groups. First group gathered the cells that were not treated with rapamycin whereas the second included those after culture with rapamycin. In every group there were cells with TSC1 or TSC2 gene silenced but also without silencing - wild type. Control group was the wild type cells, both in the treated or untreated group.
- The discussion should start with a summary of the main results obtained.
Response:
According to reviewer’s suggestion we have added the 2 sentences with the summary of the main results in to the discussion:
“The main results of our study was demonstration of miRNA dysregulation in cell lines silencing TSC1 or TSC2 genes. Incubation with rapamycin caused normalization of miR-21-3p, miR-433-3p, let-7q-3p in TSC1 group and miR-1224-3p in TSC2 group. However some of miRNA did not normalized after rapamycin treatment: miR-33a-3p, miR-29-3p in TSC1 group and miR-29a-3p in TSC2 group.”
- the link between the sentences from lines 178-180 to lines 180-183 is not clear to me.
Moreover, the reference to the Calsina et al. 2019’s work is not clear.
Response:
We have added the sentence to explain the link between the sentences from lines 178-180 to lines 180-183 and the reference to Calasina et a. 2019:
“Upregulation of some miRNA and normalization after rapamycin treatment correspond to the therapeutic response to rapamycin in TSC, however miRNAs which did not normalize may be responsible for the partial efficacy of the therapy.”
- In general, the authors should be clearer and incisive in the whole discussion section, better comparison with other authors and works, and they should add a limitation section.
Response:
According to reviewer suggestion we have rewritten the discussion section.
- The conclusions should be much more incisive and they should look at future directions mainly in relation to translational level. This paragraph: “The variability of miR-29a-3p and miR-33a-3p after rapamycin treatment and the lack of changes in their downstream regulatory targets suggests that they might be mTOR independent.” (lines 392-393, lines 40-41) is not clear.
Response:
According to reviewer suggestion we have rewritten this paragraph:
Conclusions
“The dysregulations of miRNAs found in cell lines silencing TSC1 or TSC2 genes before and after rapamycin treatment indicate their involvement in TSC pathogenesis. The autonomic miRNA (miR-29a-3p and miR-33a-3p) may represents an attractive target for further preclinical studies in TSC related tumors. Further studies are needed to confirm link between C/EBPβ and p53 transcription factors and autonomic miRNA.”
The abstract, the end of the discussion and the conclusions are the same words.
Answer:
According to reviewer suggestion we have rewritten this paragraphs:
Abstract:
In conclusion, results of our study indicate the involvement of miRNA dysregulation in the pathogenesis of TSC. Some of miRNA might be used as markers of treatment efficacy and autonomic miRNA as a target for future therapy.
Discussion:
“Therefore, these findings suggest that miRNA may play a pivotal role in TSC pathogenesis, and that some may serve as biomarkers of treatment efficacy. The autonomy of miR-29a-3p and miR-33a-3p in relation to rapamycin treatment and the lack of changes in their downstream regulatory targets suggest that they might be mTOR independent. One explanation could be that other signaling pathways are connected to mTOR by the microRNA network. The most promising crosstalk signaling pathways may involve C/EBPβ or p53 transcription factors; however, further studies are needed. Additionally miR-29a-3p and miR-33a-3p could serve as a targeted therapy for tumor associated with TSC not responding to rapamycin treatment.”
Conclusions:
“The dysregulations of miRNAs found in cell lines silencing TSC1 or TSC2 genes before and after rapamycin treatment indicate their involvement in TSC pathogenesis. The autonomic miRNA (miR-29a-3p and miR-33a-3p) may represents an attractive target for preclinical studies in TSC related tumors in future. Further studies are needed to confirm link between C/EBPβ and p53 transcription factors and autonomic miRNA.”
Minor points:
- Check for genes: they must be written in italics (for instance lines 77-78, 16, 164,165).
- Check for English errors, for instance “The variability of miR-29a-3p and miR-33a-3p after rapamycin treatment and the lack of changes in their downstream regulatory targets suggests (suggest) that they might be mTOR independent.” (lines 392-393, lines 40-41); “Another two miRNAs…” (line 195).
- There are two repetitions in the abstract section: “data” in lines 38-39.
- Check for typos, for instance, eliminate point at line 126, insert point at line 270. Check lines 204, 208-209.
- There is a difference in font size at the lines 263-268.
Response:
The errors were corrected according to reviewer suggestions (points 1-5).
In lines 163-165 transcription factors are listed, that’s why we did not write it in italics.

Round 2
Reviewer 2 Report
The authors answered to all my comments.